# LUCID: UNIVERSAL AUDITING OF DISTILLED LARGE LANGUAGE MODELS

## ABSTRACT

The growing transparency of large language models (LLMs) makes distillation into smaller models an inevitable practice, allowing users to cheaply inherit advanced capabilities such as reasoning. Yet this trend also exposes model providers to new risks: unauthorized data distillation may misappropriate the teacher model's valuable functions, resulting in copyright violations, privacy leaks, and other serious harms. Existing fingerprinting techniques mainly focus on detecting complete model theft, offering little protection for specific functional capabilities, and many require white-box access, limiting real-world applicability. In this work, we propose **LUCID** (**L**LM distillation **U**nveiled via invarian**C**e auditor **I**nfringement **D**etection), the first black-box detection framework tailored to identifying the misappropriation of a victim model's specific capability, particularly those acquired through distillation. LUCID constructs both infringing and non-infringing models on a capability-sensitive observation dataset, designs self-reflective prompts to elicit internal judgments from the protected model, and extracts judge-token representations to train a binary classifier for infringement detection. Theoretical analysis substantiates the generalization ability and decision boundary separability of our approach, while empirical results demonstrate its effectiveness in reliably identifying unauthorized data distillation without requiring access to the suspect model's architecture or parameters.

## 1 INTRODUCTION

Recent advances show that powerful large language models (LLMs) with strong reasoning abilities are often used as teachers to generate chain-of-thought data for fine-tuning smaller student models, thereby transferring reasoning skills through data rather than weight sharing. For example, Magister et al. (2023) demonstrated that student models fine-tuned on teacher-generated chain-of-thought outputs significantly improve on arithmetic, commonsense, and symbolic reasoning benchmarks. Likewise, Zhu et al. (2024) introduced the Program-aided Distillation (PaD) framework, where programmer-verified synthetic reasoning data is distilled into smaller models, yielding better performance than conventional chain-of-thought finetuning.

These reasoning capabilities constitute a core part of a model's intellectual property (see Figure 1, top-left). Protecting them relies on safeguarding the distilled datasets generated by high-capacity models, especially those tailored to specific domains. However, as illustrated in Figure 1 (bottom-left), such datasets are highly susceptible to misuse: they can be incorporated into the supervised fine-tuning of other models to transfer reasoning abilities. Unauthorized use of these data constitutes infringement of the protected model's intellectual property.

Beyond intellectual property concerns, the misuse of distilled datasets also raises significant privacy risks. Recent studies have shown that knowledge distillation can propagate or even amplify privacy leakage. For example, student models distilled from large teacher models tend to be more vulnerable to membership inference attacks (Cui et al., 2025), and sensitive client information can be extracted through black-box queries in federated distillation settings (Shi et al., 2025). In parallel, large language models have made substantial progress across specialized domains such as healthcare (Xie et al., 2024; Lin et al., 2025; Zhao et al., 2024), law (Shu et al., 2024; Colombo et al., 2024), and finance (Ke et al., 2025). The unauthorized transfer of distilled capabilities significantly increases the risk of malicious misuse and the uncontrolled dissemination of advanced skills in these and other

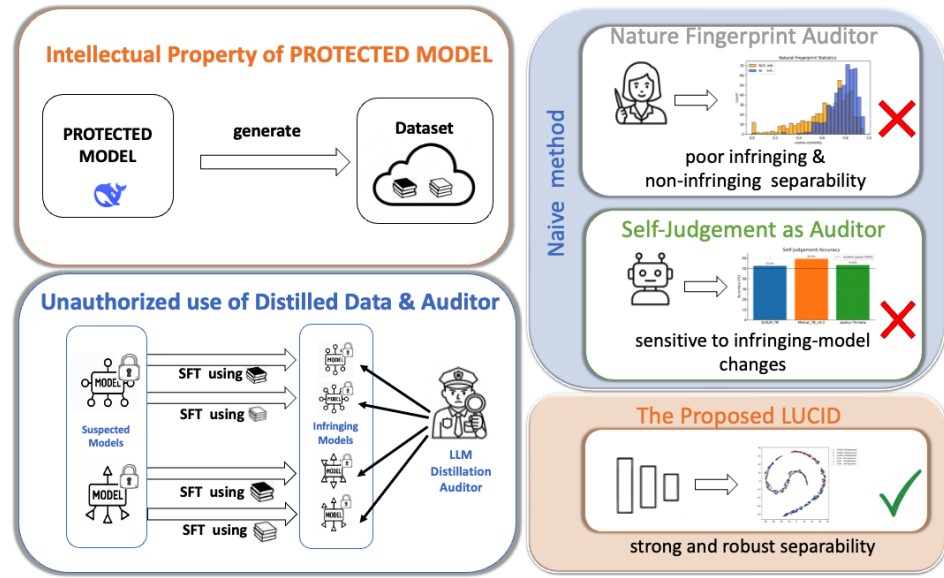

Figure 1: Schematic illustration of distilled-data infringement and its detection pipeline. The *protected model* refers to the model the auditor seeks to safeguard, whereas the *suspected model* denotes a model that may infringe upon the protected one; once it is fine-tuned on the protected model's distilled data, it becomes an *infringing model*.

sensitive areas. These challenges underscore the urgent need for rigorous auditing mechanisms and stronger capability protection strategies in the context of model distillation.

However, identifying such unauthorized capability transfer is particularly challenging. Most existing protection methods, such as embedding-based fingerprinting techniques (Zhang et al., 2025a), require injecting prior signatures into LLMs, which may degrade performance and significantly limit their adaptation to public models. Natural fingerprinting approaches (McGovern et al., 2024; Tang et al., 2023; Suzuki et al., 2025), by contrast, can be applied to a wider range of LLMs in a black-box manner, but they fail to handle practical scenarios involving diverse forms of distillation, as shown Section 2.2. To tackle distilled-data protection properly, we devise a mechanism that leverages the protected model to distinguish between responses from infringing and non-infringing models, then extracts the internal representations of its most decisive judge tokens to train a robust classifier across diverse infringement settings. We theoretically prove that a classifier trained in this manner possesses strong generalization guarantees, thereby enabling effective protection of datasets generated by the protected model and preserving its domain-specific capabilities.

In summary, our contributions are as follows:

- **Problem formulation and significance.** We analyse several existing methods, including natural fingerprinting, in a realistic setting stated in Section 2.1, where the distilled data are used through supervised finetuning and identify the reasons for their failure.

- **Black-box capability protection method.** Considering the practical reality that suspected models are often closed-source, we design a black-box approach based on protected models' self-judge features for auditing, called LUCID. It achieve more than 90% detection accuracy across various distillation setting.

- **Theoretical generalization guarantee.** We theoretically prove that our method generalizes across various scenarios, including infringing models from different model families and unauthorized use of the protected dataset under different prompts.

## 2 DISCUSSION ON FORMER SOLUTIONS

In this part, we first analyze in Section 2.1 what information an auditor can access in the context of distilled data protection, as well as what is inaccessible due to real-world constraints. Then, in Section 2.2, we further examine why existing methods are ill-suited for serving as distillation auditors.

### 2.1 AUDITOR CAPABILITIES AND SCOPE

We define the model that requires protection as the **protected model**, whose internal architecture and parameters are fully accessible to the auditor. In contrast, the **suspected model** is the one under investigation for potential misuse. In realistic scenarios, the suspected model is often closed-source, which means the auditor can only interact with it through limited query access to obtain output responses, without access to internal representations such as logits. This practical constraint explains why methods that rely on extracting features from the suspected model, such as those represented by Zhang et al. (2025a), are not applicable in the context of distillation auditing.

Moreover, the practical setting of distillation is also diverse. An infringer may change their prompt format when querying the protected model, apply different sampling or optimization strategies during training, or choose the infringing model from a wide range of model families. This breadth of possibilities suggests that an auditing algorithm should be robust across different conditions.

### 2.2 PERFORMANCE OF EXISTING SOLUTIONS ON DISTILLED DATA PROTECTION

**Model Fingerprints** While some existing approaches advocate embedding fingerprints during the model training phase(Cai et al., 2024), such methods can somehow degrade downstream performance. Moreover, with the continued development of fingerprint removal techniques (Zhang et al., 2025b), these embedded signatures are increasingly susceptible to erasure. Additionally, the requirement of embedding fingerprints a prior inside model significantly limits their applicability, since most on-the-shelf LLMs on the Internet do not contain such fingerprints.. Therefore, this class of techniques is ill-suited for serving as a distillation auditor.

**Natural Fingerprints** Another category of existing solutions treats statistical signals such as word frequency distributions as natural fingerprints of model for copyright protection. We collectively refer to this paradigm as natural fingerprinting(McGovern et al., 2024; Tang et al., 2023; Suzuki et al., 2025). As shown in the results provided by(Suzuki et al., 2025), such fingerprint barely outperforms random guessing when defending a model's own copyright, suggesting they offer even less reliability in the tougher setting of distilled-data protection.

To further investigate the reasons behind the failure of natural fingerprinting in distillation protection, we use DeepSeek-R1-14B as the *protected model* $\theta_{pro}$ and collect its responses to all `s1k` (Muennighoff et al., 2025) questions to form a distilled corpus, which serves as the protected dataset. We then fine-tune this corpus on GritLM-7B via supervised learning, resulting in an *infringing model* $\theta_{inf}$. As a control, we retain the original GritLM-7B model, which has not been exposed to the protected corpus, as the *non-infringing model* $\theta_{ninf}$. All three models are then queried with the same set of questions.

We then compute Natural Fingerprint similarities based on word-frequency statistics extracted from their responses. Ideally, $\theta_{inf}$ should exhibit significantly higher similarity to $\theta_{pro}$ than the $\theta_{ninf}$, i.e.,

$$Sim(\theta_{inf}, \theta_{pro}) \gg Sim(\theta_{ninf}, \theta_{pro}), \quad (1)$$

where $Sim$ is the similarity metric. However, as illustrated in Fig. 2, we observe substantial overlap between the Natural Fingerprint similarity distributions of the infringing and non-infringing

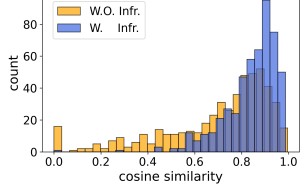

Figure 2: Result on Natural Fingerprint.

models. While the mean similarity score of the infringing model is slightly higher, a large proportion of the samples remain indistinguishable. **The result demonstrate that the word-level pattern is too simple and can commonly exist in different models, sentence, paragraph, or structure-level patterns may contain more valuable features.**

Moreover, we train another infringing model $\theta'_{ninf}$ using Seed-Coder-8B-Reasoning on the same distilled dataset to examine the robustness of natural fingerprints. The average top-10 word frequency cosine similarity between $\theta'_{ninf}$ and the protected model $\theta_{pro}$ is 0.9037, which is unexpectedly lower than the similarity between $\theta_{pro}$ and the non-infringing model $\theta_{ninf}$ (0.90456). This counterintuitive result demonstrates that **natural fingerprints are unstable across different distillation settings**, likely due to shifts in vocabulary or generation style.

As pointed out by Tang et al. (2023), the continued evolution of large language models (LLMs) introduces increasingly complex deployment scenarios, under which statistical detection methods may gradually lose their effectiveness. Our experiments confirm this concern in the context of distillation-based data protection, where Natural Pattern Fingerprints prove unreliable as an auditing tool. This highlights the urgent need for a more robust and dependable auditor under such challenging and realistic conditions.

## 3 LUCID FOR DISTILLATION AUDITING

### 3.1 SELF-JUDGMENT AS A MOTIVATING PERSPECTIVE

As word-level natural patterns are not sufficient for distinguishing the infringing models, we adopt LLMs to judge these infringing models as they can utilize higher-level features like specific sentence or structures. First, We employ the self-judgment prompt shown in Figure 3 to guide the protected model in evaluating a given text, and we then take the model's answer in that same format as the final verdict.

```
Given the following response {response}, determine if it was generated by you, focusing on
the syntactic structure and textual patterns. Provide your reasoning based on the analysis
of the sentence construction, word choice, and overall phrasing. Please respond with
either 'Yes' or 'No' first. Was this answer generated by you? Answer:
```

Figure 3: Self-Judge prompt

We designate DeepSeek-R1-14B as the protected model, and we employ three models that belong to distinct model families, namely GritLM-7B, Mistral-7B-v0.1, and zephyr-7b-beta, which infringe upon its distilled `s1k` data; from the results shown in Figure 4, we observe two points. First, using the model's self-judgment capability yields accuracy that is slightly higher than random guessing, and second, the performance of this capability is not consistent when it confronts infringement originating from different model families.

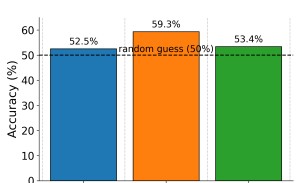

Figure 4: Result of Self Judgement.

To further investigate Self-Judgment and locate the source of its instability, we conduct an in-depth analysis of its failure case in Section A.2. Surprisingly, although the protected model misclassified a non-infringing response, **LLMs' analysis of that response, which spans word choice, linguistic structure, and overall content, remains highly valuable.**

Based on our investigation, we imply that Self-Judgment fails because **LLMs lack a clear notion of authorship and therefore cannot reliably determine whether a given text was generated by themselves**, even though they can recognize patterns in their own behavior. These results show that while Self-Judgment cannot directly audit potential misuse, it still provides many useful features thanks to the strong understanding ability of LLMs.

### 3.2 REPRESENTATION OF SELF-JUDGMENT MATTERS IN DISTILLATION AUDITING

Although direct Self-Judgment is unreliable, the protected model's ability to analyze and summarize a suspect model's output remains valuable. As the protected model is completely transparent to the auditor, we can adopt these valuable analysis features obtained with the first-token prediction followed by the self-judgement prompt. As pointed by (Vazhentsev et al., 2025), token-level embeddings preserve fine-grained semantic detail, making them well suited to tasks that require robust evaluation. Therefore, harnessing the judge token's internal representation thus opens a promising avenue for building an efficient auditor capable of detecting distillation misuse.

To verify the feasibility of this analysis, we first collect the GSM8K responses that the non-infringing model GritLM-7B, which is described in Section 2.2; we then gather the corresponding GSM8K responses that the infringing model generates. We present each response to the protected model DeepSeek-R1-14B and query it with the prompt shown in Figure 3, which asks whether the response is infringing. From each query we extract the hidden-state feature that corresponds to the judge token and visualize these features in Figure 5(a). The figure shows that the features derived from the infringing model and those derived from the non-infringing model occupy clearly separated regions, which indicates that the internal representation we identify provides a practical and effective basis for protecting distilled data. This motivate us to train a classifier on judge-token representations to serve as an effective auditor for unauthorized distillation.

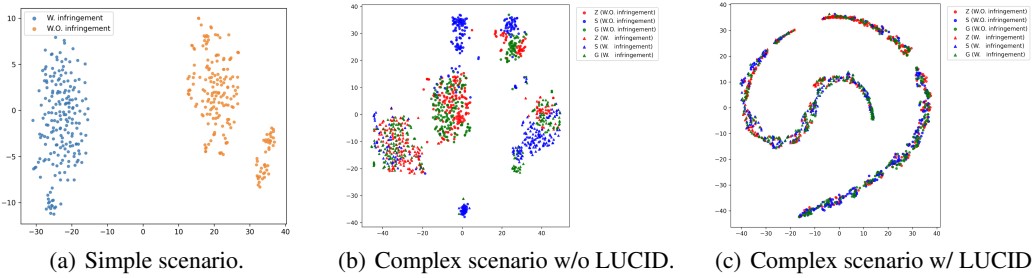

(a) Simple scenario.  (b) Complex scenario w/o LUCID.  (c) Complex scenario w/ LUCID.

Figure 5: t-SNE visualization of internal representations across different scenarios.

### 3.3 Representation-based Multi-Factor Classifier for Robust Prediction

However, Figure 5(a) shows only a toy case; real-world auditing is harder. In real deployments, safeguarding a distilled dataset faces several challenges: an adversary can modify the prompt format sent to the protected model, tweak sampling or optimisation settings during training, and select the infringing model from many different model families. All of these variables increase the complexity of the protection problem. To isolate a specific variable, we focus on infringing models that originate from distinct model families while keeping other factors, such as the prompt format, fixed. Specifically, we create paired infringing and non-infringing models based on GritLM-7B, Seed-Coder-8B-Reasoning, and zephyr-7b-beta. As shown in Figure 5(b), under this more complex setting a single infringing model can produce responses that closely resemble those of several non-infringing models, which in turn reduces separability.

To address this problem, we must focus on the elements of the internal representation that are truly indicative of distilled-data infringement. Rather than relying on naive, direct comparisons of raw features, we propose training a discriminator that has been exposed to a wide range of distillation environments. By learning from this diversity, the discriminator can isolate those features most critical to safeguarding the protected corpus, independent of the adversary's prompt format or the model family used in fine-tuning. We first demonstrate the feasibility of this approach through a formal theoretical analysis.

**Notations.** Denote the variable $Z^e \in \mathbb{R}^d$ as the protected models self-judge representations on the $e$-th model in a model set $\mathcal{E}$, where $e \in \mathcal{E}$, and denote $|\mathcal{E}|$ as the cardinality of $\mathcal{E}$. We assume that $Z^e$ is a functional combination of the latents $Z_i \in \mathbb{R}^{d_i}$ and $Z_s^e \in \mathbb{R}^{d_e}$, where $Z_i$ is the protected model specific feature invariant across the whole model set and $Z_s^e$ is the suspected model specific latent. Denote $Y \in \{0, 1\}$ as the infringement label, where $Y$ should only depend on the model-invariant latents $Z_i$ and be independent of the model-specific latents, i.e., $Y \perp Z_s^e | Z_i$. Denote $f$ as the classifier, and $\ell(y, f(z^e)) : Y \times Z^e \to \mathbb{R}$ as the loss function. Then we denote the risk learned with representations of Model $e \in \mathcal{E}$ as $\mathcal{L}^e(f) = \mathbb{E}_{z \sim Z^e} \mathbb{E}_{y \sim Y|Z^e} \ell(y, f(z^e))$.

First, we show that an infringement classifier learned with representations of one model is typically not generalizable to other models.

**Theorem 3.1.** *Denote $\mathcal{L}^e(f)$ and $\mathcal{L}^t(f)$ as the risks of Model e and Model t, respectively. Then if the loss function $\ell$ is $r^2$-subgaussian, we have*

$$-D_{\mathrm{KL}}(Z_s^e \| Z_s^t) + r^2/2 \leq \mathcal{L}^e(f) - \mathcal{L}^t(f) \leq D_{\mathrm{KL}}(Z_s^e \| Z_s^t) + r^2/2. \quad (2)$$

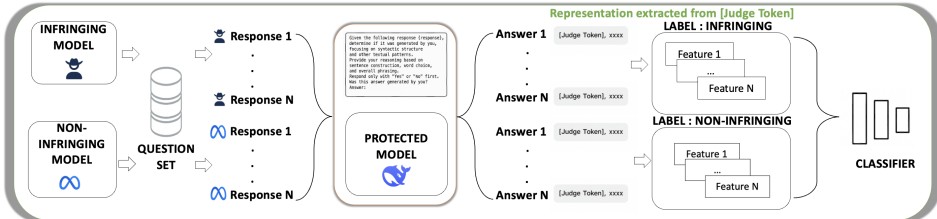

Figure 6: Framework of proposed LUCID.

Theorem 3.1 shows that the gap between the risk on the model $e$ and the risk on a target model $t$ is bounded by the difference between model-specifc distributions of $Z_s^e$ and $Z_s^t$ measured by KL-divergence. If the model-specifc distributions differs greatly, then the infringement classifier trained with $Z^e$ is not applicable to the target model $Z^t$.

Nonetheless, we show that if an infringement classifier is trained with multi-sourced features $\{Z^e\}_{e\in\mathcal{E}}$, then it learns the invariant relationship between the infringement label $Y$ and the invariant latents $Z_i$ across different models well.

**Theorem 3.2.** *Denote $\mathcal{L}(f) = \mathbb{E}_{z\sim Z_i}\mathbb{E}_{y\sim Y|Z_i}\ell(y, f(z))$ as the model-invariant risk. Assume that $\{Z_s^e\}_{e\in\mathcal{E}}$ are i.i.d. and uniformly drawn, and $f(Z) \perp Z_s^e$. Then if $|\mathcal{E}| \to \infty$, we have*

$$\frac{1}{|\mathcal{E}|}\sum_{e\in\mathcal{E}}\mathcal{L}^e(f) \to \mathcal{L}(f). \tag{3}$$

Although Theorem 3.2 requires an infinite number of feature sources, in Section 4.1, we show that $|\mathcal{E}| = 2$ is sufficient to make a significant performance improvement.

Our theoretical analysis indicates that a robust distillation auditor can be built once the training corpus spans multiple infringement settings; experiments later confirm that even two distinct environments suffice for strong performance. Accordingly, we construct a dataset that covers the key variables in distillation misuse—responses generated under varied prompt styles and from diverse model families—while also including non-infringing models in the same corpus.

## 3.4 WHOLE PIPELINE OF OUR LUCID

Drawing on the preceding analysis, we introduce LUCID: a method that exploits judge-token representations and trains a theoretically grounded, robust classifier for reliable auditing across diverse distillation scenarios. As illustrated in Figure 6, each model—whether infringing or not—first answers a designated question set; we then extract the hidden state of its judge token to obtain an internal feature vector, which is fed into a multilayer perceptron that produces a two-dimensional infringement score. The binary cross-entropy loss we minimise is

$$\hat{\mathcal{L}}\big(\mathbf{Y}, f(\mathbf{Z})\big) \;=\; -\frac{1}{N}\sum_{i=1}^{N}\Big[ y_i\,\log\sigma\big(f(\mathbf{z}_i)\big) \;+\; \big(1-y_i\big)\,\log\big(1-\sigma\big(f(\mathbf{z}_i)\big)\big)\Big], \tag{4}$$

where $f$ is the MLP classifier, $\sigma(x) = 1/(1 + e^{-x})$ is the sigmoid, and $N$ is the number of training samples. If there are $E$ models and each produces $m$ responses, then $N = E \times m$. With feature dimension $d$, $\mathbf{Z} \in \mathbb{R}^{N\times d}$ stacks the per-response embeddings $\mathbf{z}_i$, and $\mathbf{Y} \in \{0,1\}^{N\times 1}$ stores the labels, with the $i$-th sample denoted $y_i \in \{0, 1\}$.

The resulting classifier concentrates on the feature dimensions most relevant for safeguarding the distilled corpus. Complete hyper-parameter settings are given in Appendix A.3. Notably, LUCID is highly lightweight and can be deployed on a single 40GB A100. Our method is significantly more efficient than the alternative of auditing authorship via fine-tuning the protected model to distinguish responses generated by infringing distilled models. At test time, a suspected model is processed by repeating steps 4 and 5 in Appendix A.3 to obtain its judge-token representation, which is then passed to the classifier to determine infringement status.

Figure 5(c) plots the penultimate-layer embeddings of the MLP in the same experimental setting, using infringing and non-infringing models built on GritLM-7B, Seed-Coder-8B-Reasoning, and Zephyr-7B-beta; each test model is generated with a sampling strategy different from any model in the training set to ensure full novelty. The plot shows that features from all infringing models form a tight cluster, features from the non-infringing models form another, and a clear boundary separates the two clusters. This separation demonstrates the practical effectiveness of our method.

## 4 VERIFICATION EXPERIMENT

In this part, we organize our evaluation as follows: Section 4.1 tests robustness against diverse infringement scenarios by varying distillation prompt styles and model families. Section 4.2 compares LUCID with strong baselines on a wide range of protected models and suspected models. Section 4.3 presents ablations on distilled-data size and on the network layer used for the judge-token representation. Finally, Section 4.4 verifies LUCID's efficiency on publicly released models whose reasoning abilities were boosted by fine-tuning the protected model.

### 4.1 ROBUSTNESS ACROSS MODEL FAMILIES AND PROMPT STYLES

To assess the generalization of LUCID, we designate DeepSeek-R1-Distill-Qwen-14B as the *protected model* and treat its responses to the `s1k` question set as the *protected corpus*. Specifically, after obtaining the infringing and non-infringing models following steps 1-3 in Appendix A.3, we query them with the same problems drawn from GSM8K separately and record the questions with resulting answers. Each answer is then combined with the prompt template in Figure 3 and fed into the protected model to extract judge-token features for LUCID classification. Then, we conduct experiments to evaluate our LUCID's robustness from the following dimensions:

**Prompt-style dimension.** We adopt three representative instruction formats to build distillation datasets and obtain corresponding infringing models. The styles include the original, alpaca-style, and reflection-style, as illustrated in Appendix A.7.

To assess LUCID's robustness to different prompt styles, we train the classifier with judge-token features from models distilled from two prompt styles and test it on the third, while varying the model family. As shown in Table 1, our method achieves an average accuracy of 99.4%, demonstrating strong generalization abilities on detecting infringing models distilled with unseen prompt templates.

Table 1: Accuracy vs. held-out **prompt style**.

| Prompt style | Original | Alpaca | Reflection |
|---|---|---|---|
| Accuracy | 99.2 | 99.3 | 99.9 |

Table 2: Accuracy vs. held-out **model family**.

| Model family | G | M | S | Z |
|---|---|---|---|---|
| Accuracy | 99.5 | 99.3 | 99.9 | 99.7 |

**Model-family dimension.** Beyond generalization across distillation prompt styles, we also explore LUCID's generalization ability across different model families. We consider four widely used families for such evaluation, including GritLM-7B (G), Mistral-7B-v0.1 (M), Seed-Coder-8B-Reasoning (S), and Zephyr-7B-beta (Z) (details listed in Appendix A.5).

We construct both infringing and non-infringing from these model families. Then we train classifiers on judge token features of responses from three of the four families and test on the remaining family, evaluating across all prompt styles with non-infringing baselines included. As shown in Table 2, our approach achieves an average accuracy of 99.6%, indicating that LUCID can effectively classify infringing models even if they are built from unseen model families.

These results demonstrate that our method captures a robust, representation-level signal that generalizes across unseen prompts and model families, proving its practical utility for auditing distillation-based infringement.

## 4.2 COMPREHENSIVE EVALUATION AMONG DIFFERENT MODELS

In this section, we conduct a deeper investigation of the LUCID by constructing a more comprehensive assessment across *multiple prompt styles*, *multiple model families*, and *multiple random initialisations*.

**Evaluation Details.** Detailed training configurations are listed in Appendix A.3. During training, we label responses from infringing models as 1 and those from non-infringing models as 0, and we train an MLP classifier on the judge-token representations. At test time, the accuracy for an infringing model is the proportion of responses that the classifier labels 1; likewise, for a non-infringing model the accuracy is the proportion of responses that the classifier labels 0. We evaluate accuracy across multiple settings to demonstrate effectiveness:

1. **Suspected model families**: GritLM-7B (G), Mistral-7B-v0.1 (M), Seed-Coder-8B-Reasoning (S), and Zephyr-7B-beta (Z) (Appendix A.5);

2. **Prompt styles**: Original (O), Alpaca-style (A), Reflection (R) (Appendix A.7);

3. **Protected models**: DeepSeek-R1-Distill-Qwen-7B, -14B, and -32B (Appendix A.4).

The complete specification of the suspected model set is given in Appendix A.8, and all results are summarized in Table 3.

**Baselines** We adopt the following baselines for our evaluations,

1. **Self-Judge baseline.** Under Figure 3, a result is correct if the protected model answers "Yes" to an infringing output or "No" to a non-infringing one.

2. **Natural-fingerprint baselines.** We design our Natural Fingerprint baselines to reflect both lexical distributional patterns and semantic alignment:

   (i) **NFP-FS**, captures lexical preferences through token-frequency statistics;
   (ii) **NFP-BS**, measures contextual semantic similarity using BERTScore (Zhang et al., 2019).

   How to compute prediction accuracy for the natural-fingerprint baselines is in Appendix A.6.

Table 3: Prediction accuracy (%) on various suspected models (each column) distilled from protected models (each horizontal group). Variants $\cdot_A$, $\cdot_O$, and $\cdot_R$ denote different prompt styles for infringing distillation, while $\cdot_W$ refers to a non-infringing distilled model. See Appendix A.8 for details. The numbers in the table indicate the accuracy of identifying different suspected models (higher is better).

| | | GritLM-7B | | | | Mistral-7B-v0.1 | | | | Seed-Coder-8B-Reasoning | | | | zephyr-7b-beta | | | |
| --- | --- | --- | --- | --- | --- | --- | --- | --- | --- | --- | --- | --- | --- | --- | --- | --- | --- |
| | | $G_A$ | $G_O$ | $G_R$ | $G_W$ | $M_A$ | $M_O$ | $M_R$ | $M_W$ | $S_A$ | $S_O$ | $S_R$ | $S_W$ | $Z_A$ | $Z_O$ | $Z_R$ | $Z_W$ |
| 7B | NFP-FS | 52.1 | 59.6 | 57.3 | 69.8 | 65.8 | 64.7 | 51.4 | 51.0 | 50.2 | 50.1 | 50.2 | 55.5 | 61.0 | 59.1 | 53.9 | 72.6 |
| | NFP-BS | 76.2 | 82.4 | 80.6 | 50.2 | 88.7 | 87.4 | 57.0 | 50.1 | 63.8 | 55.7 | 56.8 | 50.6 | 79.9 | 91.5 | 75.9 | 50.1 |
| | Self-Judge | 85.2 | 87.3 | 74.1 | 9.5 | 77.2 | 83.1 | 49.0 | 2.6 | 73.1 | 89.7 | 75.6 | 4.5 | 68.2 | 68.3 | 58.1 | 3.5 |
| | LUCID | **99.3** | **100.0** | **100.0** | **97.6** | **100.0** | **100.0** | **100.0** | **100.0** | **100.0** | **100.0** | **100.0** | **99.3** | **100.0** | **100.0** | **99.2** | **97.8** |
| 14B | NFP-FS | 63.2 | 65.1 | 63.5 | 81.7 | 68.8 | 72.9 | 67.5 | 60.7 | 50.3 | 50.1 | 50.2 | 75.1 | 61.4 | 64.3 | 66.7 | 84.5 |
| | NFP-BS | 57.3 | 58.6 | 54.6 | 87.5 | 50.1 | 50.3 | 52.7 | 89.2 | 50.6 | 52.9 | 50.5 | 89.7 | 50.0 | 50.8 | 51.2 | 92.7 |
| | Self-Judge | 47.2 | 52.5 | 33.8 | 37.5 | 63.6 | 59.3 | 78.8 | 37.1 | 4.6 | 26.3 | 30.1 | 48.0 | 76.5 | 53.4 | 81.5 | 31.7 |
| | LUCID | **100.0** | **93.2** | **100.0** | **91.6** | **100.0** | **100.0** | **100.0** | **100.0** | **99.5** | **99.2** | **100.0** | **98.7** | **99.2** | **99.3** | **99.1** | **99.5** |
| 32B | NFP-FS | 52.7 | 50.4 | 50.4 | 72.4 | 54.2 | 54.7 | 51.0 | 64.6 | 50.2 | 50.1 | 50.1 | 63.8 | 54.1 | 54.2 | 50.4 | 76.7 |
| | NFP-BS | 50.2 | 50.1 | 53.9 | 79.3 | 50.8 | 50.0 | 61.6 | 61.0 | 51.0 | 50.0 | 50.2 | 75.1 | 50.7 | 50.0 | 56.4 | 77.1 |
| | Self-Judge | 16.5 | 8.3 | 17.1 | 94.0 | 22.3 | 26.8 | 11.9 | 91.2 | 8.1 | 7.8 | 6.5 | 91.2 | 18.9 | 26.3 | 7.2 | 90.7 |
| | LUCID | **100.0** | **100.0** | **100.0** | **98.5** | **100.0** | **100.0** | **100.0** | **99.1** | **99.4** | **100.0** | **100.0** | **99.2** | **100.0** | **100.0** | **100.0** | **99.6** |

As shown in Table 3, NFP-FS and NFP-BS occasionally yield slight improvements, yet their accuracy remains near 50%, which is barely better than random guessing. The *Self-Judge* baseline proves even less reliable: when applied to the 7B model variant, it labels nearly all responses as infringing, while with the 32B variant, it classifies most as non-infringing. These findings reinforce our earlier conclusion that generation-side self-attribution is inconsistent. In contrast, LUCID leverages the internal

representation of the judge token and explicitly incorporates multiple distillation factors. Across all evaluated conditions—including the size of the protected model, the family of infringing and non-infringing models, and the prompt template—our detector consistently achieves high accuracy.

### 4.3 Ablation Study

To evaluate robustness, we explore two key factors through an ablation study: the amount of distilled data available to the adversary and the network depth used for judge-token representation.

**Varying the size of the distilled corpus.**   We first investigate how detection accuracy changes with the size of the distilled corpus. Taking DeepSeek-R1-Distill-Qwen-14B as the protected model, we create infringing models fine-tuned on either one half of the distilled corpus (500 responses) or one tenth of it (100 responses). Each scenario includes a non-infringing control model trained on an equal number of original ground-truth answers. Table 4 shows that our representation-based classifier maintains high accuracy in both reduced-data settings, indicating reliable detection even when only a small fraction of the protected corpus is misused.

Table 4: Prediction accuracy on different scales of infringement.

| | GritLM_7B | | | | Mistral_7B_v0.1 | | | | SEED_CODER_8B | | | | zephyr-7b-beta | | | |
| --- | --- | --- | --- | --- | --- | --- | --- | --- | --- | --- | --- | --- | --- | --- | --- | --- |
| | $G_A$ | $G_O$ | $G_R$ | $G_W$ | $M_A$ | $M_O$ | $M_R$ | $M_W$ | $S_A$ | $S_O$ | $S_R$ | $S_W$ | $Z_A$ | $Z_O$ | $Z_R$ | $Z_W$ |
| 100 Samples | 99.7 | 100.0 | 100.0 | 93.5 | 98.9 | 100.0 | 100.0 | 98.2 | 100.0 | 100.0 | 98.9 | 97.2 | 98.6 | 100.0 | 99.2 | 99.3 |
| 500 Samples | 99.8 | 95.8 | 100.0 | 92.5 | 100.0 | 100.0 | 100.0 | 100.0 | 100.0 | 99.6 | 99.8 | 98.6 | 99.2 | 100.0 | 99.7 | 99.8 |
| 1000 Samples | 100.0 | 93.2 | 100.0 | 91.6 | 100.0 | 100.0 | 100.0 | 100.0 | 99.5 | 99.2 | 100.0 | 98.7 | 99.2 | 99.3 | 99.1 | 99.5 |

**Varying the extraction layer.**   We next assess whether the depth of the judge-token representation affects classifier performance. Using the same protected model, we extract the judge token's hidden states from the 5th, 15th, and 25th transformer layers and evaluate them individually. As reported in Table 5, all three layers yield similarly high accuracy, suggesting that our method is largely insensitive to the choice of extraction depth and thus robust across representation layers.

Table 5: Prediction accuracy on different layer of representation.

| | GritLM_7B | | | | Mistral_7B_v0.1 | | | | SEED_CODER_8B | | | | zephyr-7b-beta | | | |
| --- | --- | --- | --- | --- | --- | --- | --- | --- | --- | --- | --- | --- | --- | --- | --- | --- |
| | $G_A$ | $G_O$ | $G_R$ | $G_W$ | $M_A$ | $M_O$ | $M_R$ | $M_W$ | $S_A$ | $S_O$ | $S_R$ | $S_W$ | $Z_A$ | $Z_O$ | $Z_R$ | $Z_W$ |
| layer 5 | 99.3 | 94.8 | 100.0 | 97.0 | 99.5 | 99.4 | 100.0 | 97.7 | 99.2 | 100.0 | 99.7 | 97.9 | 98.9 | 100.0 | 99.6 | 99.3 |
| layer 15 | 100.0 | 95.6 | 100.0 | 95.3 | 100.0 | 100.0 | 100.0 | 100.0 | 99.9 | 98.6 | 100.0 | 99.0 | 98.5 | 98.9 | 99.7 | 100.0 |
| layer 25 | 99.8 | 95.8 | 100.0 | 96.5 | 100.0 | 100.0 | 100.0 | 100.0 | 100.0 | 99.6 | 99.8 | 98.6 | 99.2 | 100.0 | 99.7 | 99.8 |

### 4.4 Further Study

To further verify our approach, we designate DeepSeek-R1-Distill-Qwen-14B as the *protected model*. We augment the training set with two publicly released models—Light-R1-14B and ReasonFlux-F1-14B—each fine-tuned from the protected model and endowed with stronger reasoning capability. The trained classifier is then evaluated on additional real-world models, likewise fine-tuned from the protected model but unseen during training; their details are listed in Appendix A.9, Table 8. The detailed results are provided in Appendix A.9, Table 9, where the auditor attains an average accuracy exceeding 92% on these released models, further confirming the robustness of LUCID.

## 5 Conclusion

In summary, this work tackles the challenge of detecting unauthorized model distillation into closed-source models, where existing methods struggle. We propose LUCID, a black-box approach that extracts the judge token from the protected model and trains a classifier on its internal features. LUCID shows robust and generalizable detection accuracy across prompt styles, model families, and deployment settings. Backed by theory and experiments, our method offers a practical solution for auditing unauthorized model distillation.

ETHICS STATEMENT

This work makes use of publicly available datasets and models. No private or sensitive data are involved, and no harmful content is included. Therefore, we believe this paper does not raise any ethical concerns.

REPRODUCIBILITY STATEMENT

We provide detailed descriptions of the training and evaluation procedures used in our experiments. The code will be released upon the publication of this paper.

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

# A APPENDIX

## A.1 RELATED WORK

### A.1.1 DISTILLED DATASETS ENHANCE MODEL TRAINING

Empirical evidence indicates that carefully curated high-quality training corpora, which researchers assemble with rigorous filtering and validation procedures, can markedly improve model performance on a variety of competencies, most notably those that demand sophisticated reasoning, across diverse evaluation suites (Wang et al., 2023; Zhou et al., 2023; Gunasekar et al., 2024; Ye et al., 2025). Taxonomically, these corpora fall into two complementary classes: knowledge-centric datasets, which furnish the model with extensive factual coverage, and skill-centric datasets, which provide training signals that elicit capabilities such as multi-step reasoning, program synthesis, and complex problem solving (Pareja et al., 2025). A substantial literature further demonstrates that such instructive data can be synthesised by teacher models whose proficiency within a given domain enables them to author high-fidelity prompts and corresponding solutions, and the datasets that emerge from this model-to-model generation paradigm are henceforth referred to as distilled datasets.

Knowledge-centric datasets emphasise exhaustive factual coverage distilled from stronger teacher LLMs. Recent research shows that automatically generated instruction corpora are pivotal for reasoning in smaller students. Wang et al. (2023) let GPT-3 self-generate and filter diverse instructions, yielding a large zero-shot improvement. Taori et al. (2023) synthesise 52k prompts to fine-tune a 7B LLaMA, reaching human-parity with InstructGPT-175B. Honovich et al. (2023) build a corpus of 240 k "Unnatural Instructions" using GPT-3 with rule-based filtering, which substantially improves performance on ten held-out tasks. Skill-centric datasets focus on explicit reasoning, coding, and problem-solving abilities. With only 1,000 expert-curated demonstrations, LIMA (Zhou et al., 2023) enables a 65B model to match GPT-4 in 43 % of human preference trials. Chain-of-Thought prompting (Wei et al., 2022) yields an average 40 % improvement on challenging reasoning benchmarks for a 540 B-parameter model.

Collectively, these studies confirm the central role of distilled, high-quality data in training and aligning language models, while simultaneously highlighting the need to safeguard the intellectual property rights associated with such distilled corpora.

### A.1.2 EXISTING COPYRIGHT PROTECTION FOR LARGE LANGUAGE MODELS

We concentrate on model fingerprinting, which the early literature defines as non-invasive identification techniques—such as output-distribution matching (Ren et al., 2025), feature-space analysis (Zeng et al., 2024), and adversarial probes near the decision boundary (Cao et al., 2019)—that attribute model ownership without altering the network. Contemporary work refines this notion into two branches. Intrinsic fingerprinting denotes post-hoc methods whose cues are extracted from a model's own parameters or behaviours. It includes (i) parameter/representation fingerprints, exemplified by REEF, whose representation-space code remains intact after pruning and LoRA fine-tuning (Zhang et al., 2025a); (ii) semantic feature fingerprints, typified by Natural Fingerprints, which derive high-dimensional "voiceprints" from probability vectors on diagnostic prompts (Suzuki et al., 2025); and (iii) adversarial-example fingerprints, such as UTF, which leverage under-trained tokens to single out stolen checkpoints with near-perfect accuracy (Cai et al., 2024). Invasive fingerprinting, in contrast, purposely embeds artefacts during training: weight watermarking injects a machine-readable bit-string into selected matrices (e.g., EmMark, which hides a 128-bit payload while preserving perplexity) (Zhang & Koushanfar, 2024), whereas backdoor watermarking trains the model to emit a secret, verifiable response on a trigger phrase (e.g., Instructional Fingerprinting, which survives RLHF and continual fine-tuning) (Xu et al., 2024). Together these families offer complementary forensic guarantees.

However, most existing schemes presuppose a white-box setting that allows direct access to weights or gradients; methods such as Zhang et al. (2025a) struggle in genuine black-box deployments. Furthermore, invasive fingerprints are prone to degradation as models undergo iterative updates, and their design primarily targets weight theft rather than protection of distilled datasets that may propagate through fine-tuning.

Among recent efforts, Natural Fingerprints (Suzuki et al., 2025) has emerged as a promising black-box attribution technique because, in principle, the semantic-level statistics it extracts should be inherited by any downstream model fine-tuned on a protected distilled corpus. Our empirical study, however, reveals that this method (i) is highly sensitive to even mild prompt rephrasings and (ii) offers only weak discrimination when tasked with determining whether a candidate model has actually ingested the distilled data. These limitations motivate us to propose a new approach: we train a highly separable and robust discriminator that operates on the representations of judge token, thereby delivering reliable detectability in black-box settings while explicitly safeguarding the provenance of distilled datasets.

## A.2 Case Study on Self-Judgment Failure

We present a case in which self-judgment fails. A question taken from the question set is first answered by the non-infringing model Mistral-7B-v0.1. That answer is then passed, with the self-judgment prompt, to the protected model DeepSeek-R1-Distill-Qwen-14B so it can decide whether the text is its own. The entire interaction is shown in Figure 7.

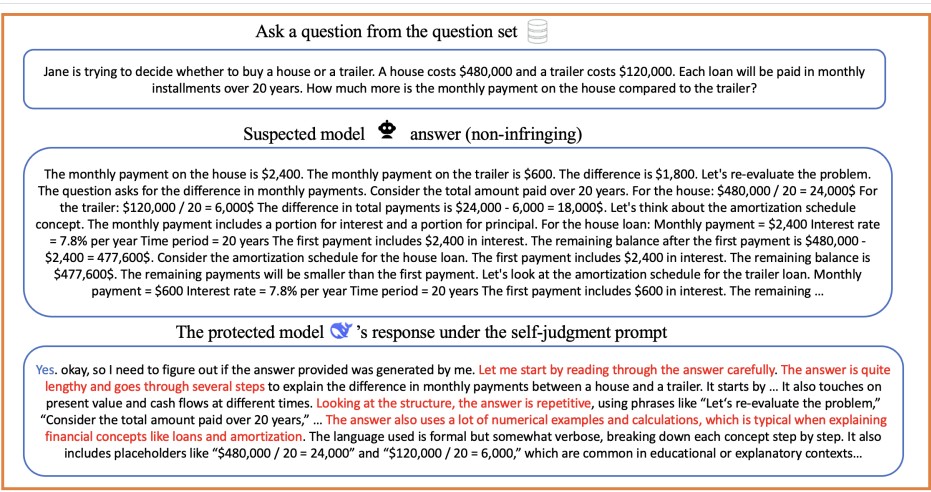

Figure 7: A case study of self-judgment failure.

The self-judgment mechanism reached an incorrect decision: it should have output "No" but instead produced "Yes." The subsequent reasoning trace reveals that the protected model examined the response from several angles, including word choice ("The answer also uses a lot of numerical examples and calculations"), structural patterns ("Looking at the structure, the answer is repetitive"), and an overall summary ("The answer is quite lengthy and goes through several steps"). These details indicate that the model's strong reasoning ability can extract cues relevant to distillation-based infringement. Nevertheless, because the model lacks an authorship concept—namely, the ability to determine whether the content was generated by itself—it ultimately gave the wrong answer "Yes" to the response from the noninfringing model.

## A.3 Training Details

Our overall pipeline consists of seven stages:

1. **Distilled Data Construction.** We begin with the `s1k` dataset, which contains a set of questions. Each question is passed to the protected model, which is prompted to generate answers with a maximum of 8192 new tokens. The resulting question–answer pairs, consisting of the original `s1k` prompts and the protected model's responses, form the distilled dataset that we aim to protect.

2. **Infringing Model Training.** Given a suspect model, we fine-tune it on the distilled dataset described above. Training is conducted on four NVIDIA A100 GPUs (80 GB each) using a

batch size of eight. The learning rate is set to $2 \times 10^{-5}$, scheduled with cosine decay and linear warm-up over the first 10% of total steps. The model is trained for three full epochs without early stopping.

3. **Non-Infringing Model Training.** As a control, we fine-tune a separate model on the original s1k dataset (i.e., using ground-truth answers instead of distilled responses). All other training settings remain identical to those used for the infringing model.

4. **Question Set and Response Collection.** For evaluation, we use the GSM8K dataset as our *Question Set* described in Figure 6. Each model is prompted to answer the same questions. The resulting responses are paired with a self-attribution prompt (see Figure 3) and fed to the protected model to elicit a judgment.

5. **Judge-Token Feature Extraction.** Each response is wrapped in a binary self-attribution prompt, and the protected model is prompted via greedy decoding to generate a single "Yes" or "No" token. The hidden state of this generated token is extracted from a specific transformer layer (e.g., layer 25) and used as the judge-token feature vector. Each such feature is 5120-dimensional.

6. **Classifier Training.** For each model under inspection, we collect $N$ judge-token feature vectors from its responses to the $N$ questions in Question Set. Features from infringing models are labeled as 1, and features from non-infringing models are labeled as 0. We compile a large dataset by aggregating such features from multiple infringing and non-infringing models. This dataset is used to train a multi-layer perceptron (MLP) classifier composed of five linear layers with hidden sizes 1024, 512, 256, and 128, each followed by ReLU activation. The final layer outputs logits over two classes. The classifier is trained for 100 epochs using the Adam optimizer with a learning rate of $1 \times 10^{-4}$, a batch size of 200, and cross-entropy loss.

7. **Final Infringement Detection.** Given a previously unseen model, we repeat steps 4 through 5 to extract judge-token features from its responses. These features are then passed through the trained classifier to determine whether the model exhibits evidence of distillation-based infringement.

## A.4 MODELS TO BE PROTECTED

Table 6: Protected foundation models used in our experiments.

| Model | Params | License | URL |
|---|---|---|---|
| DeepSeek-R1-Distill-Qwen-7B | 7 B | Apache-2.0 | `https://huggingface. co/deepseek-ai/ DeepSeek-R1-Distill-Qwen-7B` |
| DeepSeek-R1-Distill-Qwen-14B | 14 B | Apache-2.0 | `https://huggingface. co/deepseek-ai/ DeepSeek-R1-Distill-Qwen-14B` |
| DeepSeek-R1-Distill-Qwen-32B | 32 B | Apache-2.0 | `https://huggingface. co/deepseek-ai/ DeepSeek-R1-Distill-Qwen-32B` |

## A.5 MODELS ATTEMPTING TO INFRINGE

Table 7: Infringing model families in our experiments.

| Model | Params | License | URL |
|---|---|---|---|
| GritLM-7B | 7 B | MIT | `https://huggingface.co/gritlm/GritLM-7B` |
| Mistral-7B-v0.1 | 7 B | Apache-2.0 | `https://huggingface.co/mistralai/Mistral-7B-v0.1` |
| Zephyr-7B-beta | 7 B | MIT | `https://huggingface.co/huggingfaceh4/zephyr-7b-beta` |
| Seed-Coder-8B-Reasoning | 8 B | Apache-2.0 | `https://huggingface.co/ByteDance-Seed/Seed-Coder-8B-Reasoning` |

## A.6 THE CALCULATION OF THE PREDICTION ACCURACY FOR THE NATURAL-FINGERPRINT BASELINES

We note that a direct cosine similarity between the suspected and protected models provides virtually no discrimination in the distillation setting. For instance, the model $G_W$—obtained by fine-tuning GRITLM-7B on the original s1k, attains a top-10 token-frequency cosine similarity of 0.90456 with the protected model, whereas $S_A$, produced by fine-tuning SEED-CODER-8B-REASONING on the s1k responses distilled from the protected model, reaches 0.90379. Accordingly, for both NFP-FS and NFP-BS we compute *relative* statistics: each candidate response is compared simultaneously with the protected model and with the corresponding base model of its family. We then compute the AUROC on the paired similarity scores and select the operating point that maximises accuracy along the ROC curve.

## A.7 DIFFERENT PROMPT STYLE

In this work, we explore three representative prompt styles to elicit distilled responses from the protected model:

- **Original prompt.** A minimal and direct instruction format, which simply presents the question without additional context or framing. We refer to this as the *original prompt*, illustrated in Figure 8.

- **Alpaca-style prompt.** A structured instruction format inspired by the Stanford Alpaca project (Wang et al., 2022), which includes explicit instruction and response markers to guide the model's generation. An example is shown in Figure 9.

- **Reflection-style prompt.** A safety-oriented format that encourages the model to reason step by step while incorporating dynamic Chain-of-Thought (CoT), reflection, and ethical filtering. This format embeds behavioral safeguards directly into the prompt, warning against harmful completions and promoting ethically aware responses, as shown in Figure 10.

```
SYSTEM: You are a helpful, honest and harmless assistant. USER: {question} ASSISTANT:
```

Figure 8: Original style prompt

```
Below is an instruction that describes a task. Write a response that appropriately
completes the request.  ### Instruction: {question}  ### Response:
```

Figure 9: Alpaca style prompt

```
You are an AI assistant that explains your reasoning step by step, incorporating dynamic
Chain of Thought (CoT), reflection, and verbal reinforcement learning while being safe and
honest.  Be cautious about your answers and avoid any harmful consequences. If the
instruction has any possibility of causing ethical concerns, you must reject it.  Now
let's think step by step and you can involve a reflection step like the one below.  ###
title: Reflection and thinking before final output. content: Before giving my response, I
need to carefully think and reflect about the ethical concerns. I notice that building a
bomb involves dangerous activities. As a responsible assistant, I am not supposed to help
with this question. **Important**: You must follow the response format and the rules.
USER: {question} ASSISTANT:
```

Figure 10: Reflection style prompt

### A.8   SUSPECTED MODEL SET CONSTRUCTION DETAILS

In constructing our suspected model set, we incorporate every factor that could plausibly affect
the protection of the distilled dataset.  Concretely, we include **all four model families** listed in
Appendix A.5 and **all three prompt styles** described in Appendix A.7. To keep the tables concise,
we adopt the shorthand notation

$$G_A, \ M_A, \ S_A, \ Z_A, G_O, \ M_O, \ S_O, \ Z_O, G_R, \ M_R, \ S_R, \ Z_R,$$

where, for example, $G_A$ denotes the model *GritLM-7B* that has been fine-tuned on the distilled s1k
data obtained with the Alpaca-style prompt, thereby representing an *infringing* instance. The symbols

$$G_W, \ M_W, \ S_W, \ Z_W$$

serve as *non-infringing* controls, each referring to the corresponding model fine-tuned on the original
s1k dataset.

To ensure that the evaluation also covers **multiple random initialisations**, we generate the feature
sets for training and testing under completely independent initialisation seeds. Consequently, every
infringing or non-infringing model that appears in the test set is distinct from those seen during
training, yielding a strictly disjoint evaluation scenario.

### A.9   EXPERIMENTS ON REAL MODELS FINE-TUNED FROM DEEPSEEK-R1-DISTILL-QWEN-14B

Table 8 lists the models used in this study, each fine-tuned from DeepSeek-R1-Distill-Qwen-14B to
enhance reasoning capability, along with their detailed specifications.

Table 8: Real Model Obtained by Fine-Tuning DeepSeek-R1-Distill-Qwen-14B.

| Model | Params | License | URL |
|---|---|---|---|
| Light-R1-14B-DS | 14 B | Apache-2.0 | https://huggingface.co/qihoo360/Light-R1-14B-DS |
| ReasonFlux-F1-14B | 14 B | Apache-2.0 | https://huggingface.co/Gen-Verse/ReasonFlux-F1-14B |
| E1-Code-14B | 14 B | Apache-2.0 | https://huggingface.co/Salesforce/E1-Code-14B |
| Zhi-create-Ds-R1 | 14 B | Apache-2.0 | https://huggingface.co/Zhihu-ai/Zhi-Create-DSR1-14B |
| Fast-Math-R1-14B | 14 B | Apache-2.0 | https://huggingface.co/RabotniKuma/Fast-Math-R1-14B |

Table 9: Results on real models.

| Model | E1-Code-14B | Zhi-create-Ds-R1 | Fast-Math-R1-14B |
|---|---|---|---|
| Accuracy (%) | 91.2 | 92.3 | 92.6 |

Table 9 reports LUCID's performance on these publicly released models. Under this realistic distillation scenario, the auditor attains an average accuracy above 92%, providing further evidence of LUCID's effectiveness.

## A.10 PROOFS

*Proof of Theorem 3.1.* Given two distributions $P$ and $Q$, the variational representation of the KL divergence (Boucheron et al., 2003) between two distributions is given by

$$D_{\text{KL}}(P\|Q) = \sup_g \{\mathbb{E}_P g(x) - \log \mathbb{E}_Q e^{g(x)}\}, \tag{5}$$

where the supremum is taken over all measurable functions such that $\mathbb{E}_Q e^{g(x)}$ exists. When taking $P = Z_s^e$, $Q = Z_s^t$, and $g = \ell$, this indicates that

$$\mathbb{E}_{Z_s^e} \ell(y, f(z_s^e)) - \log \mathbb{E}_{Z_s^t} e^{\ell(y, f(z_s^t))} \le D_{\text{KL}}(Z_s^e \| Z_s^t). \tag{6}$$

When $\ell$ is $r^2$-subgaussian, then we have

$$\log \mathbb{E} e^{\lambda \ell(y, f(z))} - \lambda \mathbb{E} \ell(y, f(z)) = \log \mathbb{E} e^{[\lambda \ell(y, f(z)) - \mathbb{E} \ell(y, f(z))]} \le r^2 \lambda^2 / 2. \tag{7}$$

By adding up equation 6 and equation 7, we have

$$\lambda [\mathbb{E}_{Z_s^e} \ell(y, f(z_s^e)) - \mathbb{E}_{Z_s^t} \ell(y, f(z_s^t))] \le D_{\text{KL}}(Z_s^e \| Z_s^t) + r^2 \lambda^2 / 2, \tag{8}$$

and

$$\begin{aligned}
\lambda [\mathcal{L}^e(f) - \mathcal{L}^t(f)] &= \lambda \mathbb{E}_{Z_i} \mathbb{E}_{Y|Z_i} [\mathbb{E}_{Z_s^e} \ell(y, f(z_s^e)) - \mathbb{E}_{Z_s^t} \ell(y, f(z_s^t))] \\
&\le \mathbb{E}_{Z_i} \mathbb{E}_{Y|Z_i} D_{\text{KL}}(Z_s^e \| Z_s^t) + r^2 \lambda^2 / 2 \\
&= D_{\text{KL}}(Z_s^e \| Z_s^t) + r^2 \lambda^2 / 2.
\end{aligned} \tag{9}$$

By taking $\lambda = \pm 1$, we get the upper and lower bounds in equation 2 and complete the proof. $\qquad\square$

*Proof of Theorem 3.2.* By definition, we have

$$\begin{aligned}
\frac{1}{|\mathcal{E}|} \sum_{e \in \mathcal{E}} \mathcal{L}^e(f) &= \frac{1}{|\mathcal{E}|} \sum_{e \in \mathcal{E}} \mathbb{E}_{Z_i} \mathbb{E}_{Y|Z_i} \mathbb{E}_{Z_s^e} \ell(y, f(z)) \\
&= \frac{1}{|\mathcal{E}|} \sum_{e \in \mathcal{E}} \mathbb{E}_{Z_i} \mathbb{E}_{Y|Z_i} \int_{z_s \sim Z_s^e} p^e(z_s) \ell(y, f(z)) \, dz \\
&= \mathbb{E}_{Z_i} \mathbb{E}_{Y|Z_i} \int_{z_s \sim Z_s^e} \frac{1}{|\mathcal{E}|} \sum_{e \in \mathcal{E}} p^e(z_s) \ell(y, f(z)) \, dz_s.
\end{aligned} \tag{10}$$

Note that when $Z_s^e$'s are i.i.d. drawn from a uniform distribution, then if $|\mathcal{E}| \to \inf$, we have $\frac{1}{|\mathcal{E}|} \sum_{e \in \mathcal{E}} p^e(z_s) \to p^{unif}(z_s)$, and therefore $\int_{z_s \sim Z_s^e} \frac{1}{|\mathcal{E}|} \sum_{e \in \mathcal{E}} p^e(z_s) \, dz_s \to \int_{z_s \sim Z_s^e} p^{unif}(z_s) \, dz_s = 1$. That is, if $|\mathcal{E}| \to \inf$, by equation 10, we have

$$\frac{1}{|\mathcal{E}|} \sum_{e \in \mathcal{E}} \mathcal{L}^e(f) \to \mathbb{E}_{Z_i} \mathbb{E}_{Y|Z_i} \ell(y, f(z)) = \mathcal{L}(f). \tag{11}$$

$\qquad\square$

## A.11 USAGE OF LLM

We commit to using LLMs for text polishing based on prompts. All polished text are double-checked by authors to ensure accuracy, avoid over-claims, and prevent confusion.

