# OpenReview forum: "LUCID: Universal Auditing of Distilled Large Language Models"
_ICLR.cc/2026/Conference — Submitted to ICLR 2026_

### Official Review · Reviewer_H5sS · 2025-10-15

**Soundness:** 2
**Presentation:** 2
**Contribution:** 3
**Rating:** 2
**Confidence:** 4

**Summary:**

This paper introduces the problem of detecting if a model has been fine-tuned on a certain models outputs. To this end the authors introduce LUCID a method for building a classifier on top of a LLM called the protected model, to detect if a certain data set has been used to fine tune other downstream LLMs. Specifically, the protected model is prompted to make an assertion of whether it believe text was generated by itself. Rather than use its binary responses directly {yes or no}, a MLP classifier is built on top of its token representation of this answer token. This classifier is trained on a data set with two classes. The first class correspond to responses from models fine-tuned on a data set produced by the protected model. The other class correspond to responses from models fine-tuned on a different data set specifically the ground truth response of s1k. The authors verify this classifier has high accuracy on detecting these two classes apart even. This result hold across different prompting strategies and down stream architecture.

**Strengths:**

The problem tackled is relevant and to the best of my knowledge the approach is novel.
The results presented are compelling.

**Weaknesses:**

While I believe this line of research is interesting and the results presented so far are compelling I do not believe the work is ready for publication at this time, due to its poor presentation and experimental design having some fairly major flaws (over simplifications).

*Written Quality*
- Large parts are missing technical details
- The Natural Fingerprints section is very vague introducing concepts such as "similarity metric" without the details.
- The theory in Section 3.3 is also vague, rushed and difficult to follow. Making vague statements like "functional combination of the latents", without more detail. The notation in the proofs of these results is also inconsistent changing between lines, see expectations in appendix A.10 equations 6-7-8. Sadly this vagueness makes it very hard to follow or know what assumptions are being made here, so its hard to comment in more detail. Until the presentation of this section is greatly improved I do not think this section adds anything to the paper. Its also currently hard to tell why this theory its relevant method presented.
- There are also sentences that don't make sense lines 216-217 lines 338-339.
- optimiser details missing in A3 step 2.
- Figure 5 is too small


*Experiment*
The main problem with the experiments is that they verify that the binary classifier can detect which dataset the various down stream models where trained on, from a set of exactly two possible options. Not whether a specific models outputs were used for distillation.

In other words the set of possible options compared against is unrealistic and overly simplistic. In the real world when performing and audit of a model the options for where the models were trained on are not binary. There are countless options for what data the down stream models could have been fine tuned on including but not limited to:
1. being distilled from another model's outputs (not the protected model)
2. being distilled on multiple model's outputs
3. being distilled on a different data set generated by the protected model
4. being fine-tuned on some other unknown data sets
5. being fine-tuned on a mix of data sets
6. not being fine-tuned at all

However, the assumption made in the paper is a model was trained on this one specific data set generated by the protected model, or they were trained on this exact other data set. This is sadly a massive over simplification. Thus the results presented make it difficult to know how useful LUCID would be in practice, and why I believe this work is not yet ready for publication.

I encourage the authors to perform experiments with a more realistic set up, and resubmit if their method still offers compelling results.

**Questions:**

Could you explain what you are trying to show in the Theorem 3.1 and 3.2 and why this is relevant to LUCID?

---

### Official Review · Reviewer_zBNw · 2025-10-31

**Soundness:** 3
**Presentation:** 2
**Contribution:** 2
**Rating:** 2
**Confidence:** 4

**Summary:**

This paper studies the problem of auditing unauthorized distillation of large language models (LLMs) in a black-box setting. It introduces LUCID, a framework that leverages the protected model’s self-judgment token representations to train a binary classifier for detecting capability infringement. The method is supported by theoretical analysis of generalization guarantees and evaluated empirically across different model families, prompt styles, and corpus sizes. Experimental results show high detection accuracy and robustness under varied distillation conditions.

**Strengths:**

1. Model fingerprinting detection are very popular these days, and extending to distillation-specific auditing is a meaningful topic.

2. Prior works mostly assume white-box or watermark access, black-box detection for capability misuse is valuable.

3. Implementation completeness. The authors conducted experiments across multiple model families (GritLM, SeedCoder, Zephyr, Mistral) and prompt styles (original, alpaca, reflection), with strong empirical results and clear reproducibility.

**Weaknesses:**

1. According to sec 2.1, the paper defines a “black-box auditing” setting but simultaneously assumes full access to the protected model’s internal parameters, which corresponds to a one-sided black-box scenario. This distinction may be confusing.

2. Accroding to the experiment setting in A.3, the experimental validation mainly tests whether a model has been fine-tuned on a specific distilled dataset, which may not fully capture the broader notion of capability infringement that the paper aims to address.

3. Theorems 3.1 and 3.2 mainly restate standard results from domain generalization, showing that combining features from multiple sources can reduce risk divergence across models. However, the analysis does not clearly define what distillation infringement means or explain how the theoretical quantities (e.g., model-specific distributions $ Z_s^e  $) relate to the actual auditing process in practice.

4. The use of responses' internal representations from the protected model as discriminative features has been explored in prior work on model fingerprintint and watermarking. So, section 3.3 need more principled explanation of why these features uniquely enable capability-level infringement detection.

5. Since the scope of the paper is model infringing detection, but the experimental settings are largely controlled, with infringing and non-infringing models fine-tuned on the same dataset. This setup seems may not fully capture realistic unauthorized distillation scenarios.

6. The experiments use only the DeepSeek-R1-Distill-Qwen series as the protected model. It would strengthen the work to include additional reasoning-capable models as protected sources to demonstrate generality.

7. The text in Figures 4–6, including legends, is too small to read clearly, and the figure captions are relatively brief. Improving readability and providing more descriptive captions would enhance presentation quality.

**Questions:**

Please refer to the weakness section

**Details Of Ethics Concerns:**

No ethics concerns

---

### Official Review · Reviewer_AnKB · 2025-10-31

**Soundness:** 1
**Presentation:** 2
**Contribution:** 1
**Rating:** 2
**Confidence:** 4

**Summary:**

This paper propose LUCID to identifying the misappropriation of a victim model’s specific capability, particularly those acquired through distillation. LUCID constructs both infringing and non-infringing models on a capability-sensitive observation dataset, designs prompts to elicit internal judgments from the protected model, and extracts judge-token representations to train a binary classifier for infringement detection.

**Strengths:**

- The issue of IP infringement caused by model distillation is a noteworthy problem to address.

**Weaknesses:**

- There are inaccuracies in the descriptions. For instance (Lines 83-85), embedding-based fingerprinting methods do not necessarily require injecting prior signatures into the LLMs.
- Crucial experimental details are missing. For example:
  - In the discussion of "Natural Fingerprints" (Section 2.2), it is unclear what data was used for model queries (Lines 148-149).
  - It is not specified what the cosine similarity in Figure 2 is calculated between.
- The conclusions drawn from preliminary experiments appear biased.
  - In Figure 2, although an overlap exists, the statistical distributions between the two groups are clearly distinguishable.
  - For a binary classification problem, Figure 4 is insufficient to support the conclusion drawn in Lines 194-197.
  - In Figure 5(b), the circles and triangles are visibly and significantly separable even in the 2D t-SNE plot. This separation seems independent of the LUCID method, thus failing to support the method's effectiveness.
- The theoretical justification is not solid, and the underlying assumptions are questionable. The decomposition of the $Z^e$ representation into invariant and specific features (Lines 258-260) is overly idealistic and unrealistic, especially considering the entangled nature of features. Inferences based on this strong assumption are unreliable.
- Unreasonable Methodological Framework:
  - As shown in Figure 6, LUCID essentially aims to distinguish between two models fine-tuned on different answer datasets. If the two models have different base architectures, they are inherently distinguishable. If they share the same base model, the task is simply to distinguish between two different fine-tuning datasets, which is also a solvable problem. Therefore, the subsequent Self-Judge Prompt and Representation Classifier seem meaningless, as any model could likely perform this distinction, not just the specific "protected model."
  - In the experimental setup, a "Non-Infringing Model" is created via fine-tuning (Lines 761-763). This is an unnecessary step that artificially creates a binary classification task. In practice, any model other than the "Infringing Model" (e.g., the base model itself) could be considered "non-infringing."
- Insufficient Experimentation:
  - The method is claimed to be black-box, yet it is not validated on any truly black-box models.
  - The model sizes used (7B/14B) are too small. These models have limited capabilities, and the distinguishability of features might decrease significantly in more powerful models.
  - Only one category of "protected models" is used, and no cross-validation is performed.
  - Only one type of distillation data and one type of detection data are used, which makes the results unconvincing and lack generalizability.
  - There is no comparison with other established methods. The "baselines" mentioned in Lines 398-407 are functionally equivalent to the preliminary experimental setup, not proper benchmarks.
- The method has limited practical value.
  - ① Although Lines 211-212 state, "As the protected model is completely transparent to the auditor," it is highly questionable whether the method is applicable to truly closed-source models. The paper fails to provide a compelling incentive for closed-source model owners to adopt this method.
  - ② The scenario validated in the paper is too naive. Real-world distillation is far more complex, potentially involving not just fine-tuning on a single distilled dataset but also mixing multi-source data, data from multiple models, and other advanced techniques.
- Several typos are present. For example, the abbreviation "LLM" should be used after "large language models" is introduced. There is a double period in Line 135, and there are multiple instances of incorrectly used semicolons.
- Some figures are unclear (e.g., Figure 1).

**Questions:**

See weaknesses.

---

### Official Review · Reviewer_669R · 2025-11-01

**Soundness:** 2
**Presentation:** 3
**Contribution:** 2
**Rating:** 4
**Confidence:** 4

**Summary:**

**Summary**

This paper introduces a novel method named LUCID for determining whether a model has been distilled from another protected model. The authors first review existing approaches, including Model Fingerprints, Natural Fingerprints, and Self-Judgment. They observe that although Self-Judgment itself is unreliable, the hidden representations produced during the self-judgment process contain rich and discriminative information. Building upon this insight, the authors incorporate a Multi-Layer Perceptron (MLP) to effectively utilize these hidden representations, leading to the proposed LUCID framework. Experimental results show that LUCID achieves over 97% accuracy across different prompt styles and various sizes of protected models, representing a significant improvement over both Natural Fingerprints and Self-Judgment baselines. The authors further examine the robustness of LUCID by testing on infringing models with different levels of distillation and by extracting judge-token features from different Transformer layers. The results indicate that accuracy is only slightly affected and remains consistently high, demonstrating that LUCID is a robust and reliable method for detecting whether a model has been distilled from a protected one.

**Strengths:**

- This paper presents a novel approach for determining whether one model has been distilled from another. The proposed method introduces a new perspective by 	leveraging hidden-state features of language models to capture similarity patterns between model outputs.
- The proposed LUCID framework achieves substantial improvements over previous detection methods, with accuracy gains of around 30% in most evaluated settings.
- According to the experimental results, LUCID demonstrates strong robustness: even when 	experimental conditions are altered or new model families are introduced, the method consistently maintains high detection  accuracy.

**Weaknesses:**

- The paper does not discuss how the size of the infringing model may affect the performance of LUCID. Larger infringing models might exhibit weaker traces of the protected model in their outputs, potentially reducing detection accuracy.
- The paper does not specify how many independent experiments were conducted or whether the dataset used was sufficiently large to ensure statistical reliability.
- In the ablation study “Varying the size of the distilled corpus”, the authors did not include a Natural Fingerprint (NFP) baseline for comparison. Without this, it is unclear whether the robustness originates from LUCID itself or simply from small differences between infringing models trained with varying corpus sizes.
- The paper does not explore what would happen if the roles of the protected and infringing models were reversed, i.e., treating the infringing model as the protected one and vice versa.
- The paper also does not address cases where both the protected and infringing models are distilled from a third model, it is unclear how LUCID would classify such relationships, or whether it might incorrectly identify the infringing model as being derived from the protected one.
- The current method can only determine whether an infringing model has been distilled from a specific protected model. It would be interesting to explore whether LUCID could be extended to a more general detection framework, for example, by feeding the outputs of two models (a potentially distilled model and a suspected infringing model) jointly into an LLM, and then using an MLP classifier to determine whether their outputs exhibit similarity indicative of a distillation relationship.

**Questions:**

- Could the authors conduct additional experiments on larger infringing models to verify whether LUCID remains effective when the infringing models have greater capacity or reduced dependence on the protected model’s features?
- The paper presents several quantitative results across different settings (Tables 1–5), but it does not mention how many independent runs were performed for each experiment. Clarifying whether the results are based on single runs or averaged over multiple repetitions would improve the reproducibility and reliability of the findings.
- In the ablation study “Varying the size of the distilled corpus,” could the authors include an NFP (Natural Fingerprint) baseline or other supporting evidence to demonstrate that the similarity between models varies with distillation strength? This would strengthen the argument that LUCID’s robustness comes from the method itself rather than from small inherent differences between the models.
- Could the authors conduct an additional experiment by reversing the roles of the protected and infringing models (treating the infringing model as the protected one and vice versa) to observe how LUCID and other baselines behave in this setting?
- Could the authors also evaluate the case where both the protected and infringing models are distilled from a third model, to test how LUCID behaves in such hierarchical or shared-distillation scenarios?
- Finally, have the authors considered training a general-purpose LLM as a universal feature extractor, combined with a shared MLP classifier, to judge whether two models’ outputs exhibit similarity indicative of a distillation relationship? Such an extension could potentially make LUCID applicable to broader model auditing tasks beyond pairwise protected/infringing setups.

---

### Meta-Review · Area_Chair_X9gC · 2026-01-06

**Summary:**

This paper proposes a framework to detect whether an LLM has used generated data from other LLMs (via distillation) for its training.

The authors have not engaged in rebuttal, and the reviews have positioned this paper significantly under the acceptance threshold, so I recommend not accepting this paper.

**Reviewer Concerns:**

The authors have not engaged in rebuttal, and the reviews have positioned this paper significantly under the acceptance threshold, so I recommend not accepting this paper.

**Reviewer Scores:**

The authors have not engaged in rebuttal, and the reviews have positioned this paper significantly under the acceptance threshold, so I recommend not accepting this paper.

---

### Decision · Program_Chairs · 2026-01-26

Reject